# Learning Efficient Object Detection Models with Knowledge Distillation

**Guobin Chen**[1,2]   **Wongun Choi**[1]   **Xiang Yu**[1]   **Tony Han**[2]   **Manmohan Chandraker**[1,3]

[1]NEC Labs America     [2]University of Missouri     [3]University of California, San Diego

## Abstract

Despite significant accuracy improvement in convolutional neural networks (CNN) based object detectors, they often require prohibitive runtimes to process an image for real-time applications. State-of-the-art models often use very deep networks with a large number of floating point operations. Efforts such as model compression learn compact models with fewer number of parameters, but with much reduced accuracy. In this work, we propose a new framework to learn compact and fast object detection networks with improved accuracy using knowledge distillation [20] and hint learning [34]. Although knowledge distillation has demonstrated excellent improvements for simpler classification setups, the complexity of detection poses new challenges in the form of regression, region proposals and less voluminous labels. We address this through several innovations such as a weighted cross-entropy loss to address class imbalance, a teacher bounded loss to handle the regression component and adaptation layers to better learn from intermediate teacher distributions. We conduct comprehensive empirical evaluation with different distillation configurations over multiple datasets including PASCAL, KITTI, ILSVRC and MS-COCO. Our results show consistent improvement in accuracy-speed trade-offs for modern multi-class detection models.

## 1  Introduction

Recent years have seen tremendous increase in the accuracy of object detection, relying on deep convolutional neural networks (CNNs). This has made visual object detection an attractive possibility for domains ranging from surveillance to autonomous driving. However, speed is a key requirement in many applications, which fundamentally contends with demands on accuracy. Thus, while advances in object detection have relied on increasingly deeper architectures, they are associated with an increase in computational expense at runtime. But it is also known that deep neural networks are over-parameterized to aid generalization. Thus, to achieve faster speeds, some prior works explore new structures such as fully convolutional networks, or lightweight models with fewer channels and small filters [22, 25]. While impressive speedups are obtained, they are still far from real-time, with careful redesign and tuning necessary for further improvements.

Deeper networks tend to have better performance under proper training, since they have ample network capacity. Tasks such as object detection for a few categories might not necessarily need that model capacity. In that direction, several works in image classification use *model compression*, whereby weights in each layer are decomposed, followed by layer-wise reconstruction or fine-tuning to recover some of the accuracy [9, 26, 41, 42]. This results in significant speed-ups, but there is often a gap between the accuracies of original and compressed models, which is especially large when using compressed models for more complex problems such as object detection. On the other hand, seminal works on *knowledge distillation* show that a shallow or compressed model trained to mimic the behavior of a deeper or more complex model can recover some or all of the accuracy

drop [3, 20, 34]. However, those results are shown only for problems such as classification, using simpler networks without strong regularization such as dropout.

Applying distillation techniques to multi-class object detection, in contrast to image classification, is challenging for several reasons. First, the performance of detection models suffers more degradation with compression, since detection labels are more expensive and thereby, usually less voluminous. Second, knowledge distillation is proposed for classification assuming each class is equally important, whereas that is not the case for detection where the background class is far more prevalent. Third, detection is a more complex task that combines elements of both classification and bounding box regression. Finally, an added challenge is that we focus on transferring knowledge within the same domain (images of the same dataset) with no additional data or labels, as opposed other works that might rely on data from other domains (such as high-quality and low-quality image domains, or image and depth domains).

To address the above challenges, we propose a method to train fast models for object detection with knowledge distillation. Our contributions are four-fold:

- We propose an end-to-end trainable framework for learning compact *multi-class object detection* models through knowledge distillation (Section 3.1). To the best of our knowledge, this is the first successful demonstration of knowledge distillation for the multi-class object detection problem.
- We propose new losses that effectively address the aforementioned challenges. In particular, we propose a *weighted cross entropy loss* for classification that accounts for the imbalance in the impact of misclassification for background class as opposed to object classes (Section 3.2), a *teacher bounded regression loss* for knowledge distillation (Section 3.3) and *adaptation layers for hint learning* that allows the student to better learn from the distribution of neurons in intermediate layers of the teacher (Section 3.4).
- We perform comprehensive empirical evaluation using multiple large-scale public benchmarks. Our study demonstrates the positive impact of each of the above novel design choices, resulting in significant improvement in object detection accuracy using compressed fast networks, consistently across all benchmarks (Sections 4.1 – 4.3).
- We present insights into the behavior of our framework by relating it to the generalization and under-fitting problems (Section 4.4).

## 2   Related Works

**CNNs for Detection.** Deformable Part Model (DPM) [14] was the dominant detection framework before the widespread use of Convolutional Neural Networks (CNNs). Following the success of CNNs in image classification [27], Girshick et al. proposed RCNN [24] that uses CNN features to replace handcrafted ones. Subsequently, many CNN based object detection methods have been proposed, such as Spatial Pyramid Pooling (SPP) [19], Fast R-CNN [13], Faster-RCNN [32] and R-FCN [29], that unify various steps in object detection into an end-to-end multi-category framework.

**Model Compression.** CNNs are expensive in terms of computation and memory. Very deep networks with many convolutional layers are preferred for accuracy, while shallower networks are also widely used where efficiency is important. Model compression in deep networks is a viable approach to speed up runtime while preserving accuracy. Denil et al. [9] demonstrate that neural networks are often over-parametrized and removing redundancy is possible. Subsequently, various methods [5, 7, 10, 15, 17, 30] have been proposed to accelerate the fully connected layer. Several methods based on low-rank decomposition of the convolutional kernel tensor [10, 23, 28] are also proposed to speed up convolutional layers. To compress the whole network, Zhang et al. [41, 42] present an algorithm using asymmetric decomposition and additional fine-tuning. In similar spirit, Kim et al. [26] propose one-shot whole network compression that achieves around 1.8 times improvement in runtime without significant drop in accuracy. We will use methods presented in [26] in our experiments. Besides, a pruning based approach has been proposed [18] but it is challenging to achieve runtime speed-up with a conventional GPU implementation. Additionally, both weights and input activations can be the quantized( [18]) and binarized ( [21, 31]) to lower the computationally expensive.

**Knowledge Distillation.** Knowledge distillation is another approach to retain accuracy with model compression. Bucila et al. [3] propose an algorithm to train a single neural network by mimicking the output of an ensemble of models. Ba and Caruana [2] adopt the idea of [3] to compress deep

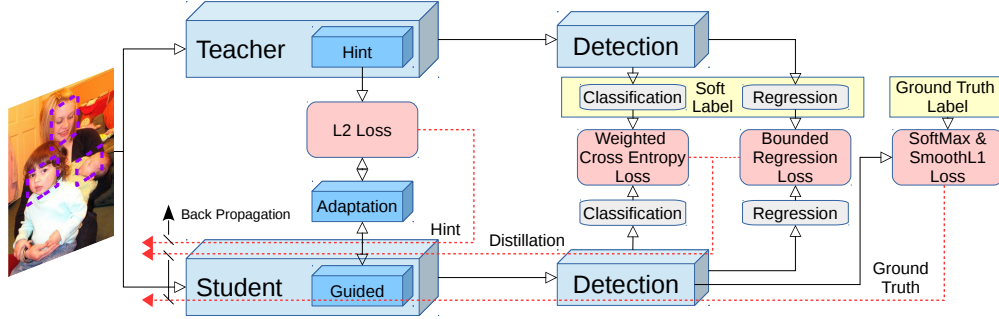

Figure 1: The proposed learning skeme on visual object detection task using Faster-RCNN, which mainly consists of region proposal network (RPN) and region classification network(RCN). The two networks both use multi-task loss to jointly learn the classifier and bounding-box regressor. We employ the final output of the teacher's RPN and RCN as the distillation targets, and apply the intermediate layer outputs as hint. Red arrows indicate the backpropagation pathways.

networks into shallower but wider ones, where the compressed model mimics the 'logits'. Hinton et al. [20] propose knowledge distillation as a more general case of [3], which applies the prediction of the teacher model as a 'soft label', further proposing temperature cross entropy loss instead of L2 loss. Romero et al. [34] introduce a two-stage strategy to train deep networks. In their method, the teacher's middle layer provides 'hint' to guide the training of the student model.

Other researchers [16, 38] explore distillation for transferring knowledge between different domains, such as high-quality and low-quality images, or RGB and depth images. In a draft manuscript concurrent with our work, Shen et al. [36] consider the effect of distillation and hint frameworks in learning a compact object detection model. However, they formulate the detection problem as a binary classification task applied to pedestrians, which might not scale well to the more general multi-category object detection setup. Unlike theirs, our method is designed for multi-category object detection. Further, while they use external region proposals, we demonstrate distillation and hint learning for both the region proposal and classification components of a modern end-to-end object detection framework [32].

# 3 Method

In this work, we adopt the Faster-RCNN [32] as the object detection framework. Faster-RCNN is composed of three modules: 1) A shared feature extraction through convolutional layers, 2) a region proposal network (RPN) that generates object proposals, and 3) a classification and regression network (RCN) that returns the detection score as well as a spatial adjustment vector for each object proposal. Both the RCN and RPN use the output of 1) as features, RCN also takes the result of RPN as input. In order to achieve highly accurate object detection results, it is critical to learn strong models for all the three components.

## 3.1 Overall Structure

We learn strong but efficient student object detectors by using the knowledge of a high capacity teacher detection network for all the three components. Our overall learning framework is illustrated in Figure 1. First, we adopt the hint based learning [34] (Sec.3.4) that encourages the feature representation of a student network is similar to that of the teacher network. Second, we learn stronger classification modules in both RPN and RCN using the knowledge distillation framework [3,20]. In order to handle severe category imbalance issue in object detection, we apply weighted cross entropy loss for the distillation framework. Finally, we transfer the teacher's regression output as a form of upper bound, that is, if the student's regression output is better than that of teacher, no additional loss is applied.

Our overall learning objective can be written as follows:

$$L_{RCN} = \frac{1}{N}\sum_i L_{cls}^{RCN} + \lambda\frac{1}{N}\sum_j L_{reg}^{RCN}$$

$$L_{RPN} = \frac{1}{M}\sum_i L_{cls}^{RPN} + \lambda\frac{1}{M}\sum_j L_{reg}^{RPN}$$

$$L = L_{RPN} + L_{RCN} + \gamma \boldsymbol{L_{Hint}} \tag{1}$$

where N is the batch-size for RCN and $M$ for RPN. Here, $L_{cls}$ denotes the classifier loss function that combines the hard softmax loss using the ground truth labels and the soft knowledge distillation loss [20] of (2). Further, $L_{reg}$ is the bounding box regression loss that combines smoothed L1 loss [13] and our newly proposed teacher bounded L2 regression loss of (4). Finally, $L_{hint}$ denotes the hint based loss function that encourages the student to mimic the teacher's feature response, expressed as (6). In the above, $\lambda$ and $\gamma$ are hyper-parameters to control the balance between different losses. We fix them to be 1 and 0.5, respectively, throughout the experiments.

## 3.2 Knowledge Distillation for Classification with Imbalanced Classes

Conventional use of knowledge distillation has been proposed for training classification networks, where predictions of a teacher network are used to guide the training of a student model. Suppose we have dataset $\{x_i, y_i\}$, $i = 1, 2, ..., n$ where $x_i \in \mathfrak{I}$ is the input image and $y_i \in \mathfrak{Y}$ is its class label. Let $t$ be the teacher model, with $P_t = softmax(\frac{Z_t}{T})$ its prediction and $Z_t$ the final score output. Here, $T$ is a temperature parameter (normally set to 1). Similarly, one can define $P_s = softmax(\frac{Z_s}{T})$ for the student network $s$. The student $s$ is trained to optimize the following loss function:

$$L_{cls} = \mu L_{hard}(P_s, y) + (1 - \mu)\boldsymbol{L_{soft}(P_s, P_t)} \tag{2}$$

where $L_{hard}$ is the hard loss using ground truth labels used by Faster-RCNN, $L_{soft}$ is the soft loss using teacher's prediction and $\mu$ is the parameter to balance the hard and soft losses. It is known that a deep teacher can better fit to the training data and perform better in test scenarios. The soft labels contain information about the relationship between different classes as discovered by teacher. By learning from soft labels, the student network inherits such hidden information.

In [20], both hard and soft losses are the cross entropy losses. But unlike simpler classification problems, the detection problem needs to deal with a severe imbalance across different categories, that is, the background dominates. In image classification, the only possible errors are misclassifications between 'foreground' categories. In object detection, however, failing to discriminate between background and foreground can dominate the error, while the frequency of having misclassification between foreground categories is relatively rare. To address this, we adopt class-weighted cross entropy as the distillation loss:

$$L_{soft}(P_s, P_t) = -\sum w_c P_t \log P_s \tag{3}$$

where we use a larger weight for the background class and a relatively small weight for other classes. For example, we use $w_0 = 1.5$ for the background class and $w_i = 1$ for all the others in experiments on the PASCAL dataset.

When $P_t$ is very similar to the hard label, with probability for one class very close to 1 and most others very close to 0, the temperature parameter $T$ is introduced to soften the output. Using higher temperature will force $t$ to produce softer labels so that the classes with near-zero probabilities will not be ignored by the cost function. This is especially pertinent to simpler tasks, such as classification on small datasets like MNIST. But for harder problems where the prediction error is already high, a larger value of $T$ introduces more noise which is detrimental to learning. Thus, lower values of $T$ are used in [20] for classification on larger datasets. For even harder problems such as object detection, we find using no temperature parameter at all (equivalent to $T = 1$) in the distillation loss works the best in practice (see supplementary material for an empirical study).

## 3.3 Knowledge Distillation for Regression with Teacher Bounds

In addition to the classification layer, most modern CNN based object detectors [26, 29, 32, 33] also use bounding-box regression to adjust the location and size of the input proposals. Often, learning a

good regression model is critical to ensure good object detection accuracy [13]. Unlike distillation for discrete categories, the teacher's regression outputs can provide very wrong guidance toward the student model, since the real valued regression outputs are unbounded. In addition, the teacher may provide regression direction that is contradictory to the ground truth direction. Thus, instead of using the teacher's regression output directly as a target, we exploit it as an upper bound for the student to achieve. The student's regression vector should be as close to the ground truth label as possible in general, but once the quality of the student surpasses that of the teacher with a certain margin, we do not provide additional loss for the student. We call this the *teacher bounded regression loss*, $L_b$, which is used to formulate the regression loss, $L_{reg}$, as follows:

$$L_b(R_s, R_t, y) = \begin{cases} \|R_s - y\|_2^2, & \text{if } \|R_s - y\|_2^2 + m > \|R_t - y\|_2^2 \\ 0, & \text{otherwise} \end{cases}$$
$$L_{reg} = L_{sL1}(R_s, y_{reg}) + \nu L_b(R_s, R_t, y_{reg}), \qquad (4)$$

where $m$ is a margin, $y_{reg}$ denotes the regression ground truth label, $R_s$ is the regression output of the student network, $R_t$ is the prediction of teacher network and $\nu$ is a weight parameter (set as $0.5$ in our experiments). Here, $L_{sL1}$ is the smooth L1 loss as in [13]. The teacher bounded regression loss $L_b$ only penalizes the network when the error of the student is larger than that of the teacher. Note that although we use L2 loss inside $L_b$, any other regression loss such as L1 and smoothed L1 can be combined with $L_b$. Our combined loss encourages the student to be close to or better than teacher in terms of regression, but does not push the student too much once it reaches the teacher's performance.

### 3.4 Hint Learning with Feature Adaptation

Distillation transfers knowledge using only the final output. In [34], Romero et al. demonstrate that using the intermediate representation of the teacher as *hint* can help the training process and improve the final performance of the student. They use the L2 distance between feature vectors $V$ and $Z$:

$$L_{Hint}(V, Z) = \|V - Z\|_2^2 \qquad (5)$$

where Z represent the intermediate layer we selected as hint in the teacher network and V represent the output of the guided layer in the student network. We also evaluate the L1 loss:

$$L_{Hint}(V, Z) = \|V - Z\|_1 \qquad (6)$$

While applying hint learning, it is required that the number of neurons (channels, width and height) should be the same between corresponding layers in the teacher and student. In order to match the number of channels in the hint and guided layers, we add an adaptation after the guided layer whose output size is the same as the hint layer. The adaptation layer matches the scale of neuron to make the norm of feature in student close to teacher's. A fully connected layer is used as adaptation layer when both hint and guided layers are also fully connected layers. When the hint and guided layers are convolutional layers, we use $1 \times 1$ convolutions to save memory. Interestingly, we find that having an adaptation layer is important to achieve effective knowledge transferring even when the number of channels in the hint and guided layers are the same (see Sec. 4.3). The adaptation layer can also match the difference when the norms of features in hint and guided layers are different. When the hint or guided layer is convolutional and the resolution of hint and guided layers differs (for examples, VGG16 and AlexNet), we follow the padding trick introduced in [16] to match the number of outputs.

## 4 Experiments

In this section, we first introduce teacher and student CNN models and datasets that are used in the experiments. The overall results on various datasets are shown in Sec.4.1. We apply our methods to smaller networks and lower quality inputs in Sec.4.2. Sec.4.3 describes ablation studies for three different components, namely classification/regression, distillation and hint learning. Insights obtained for distillation and hint learning are discussed in Sec.4.4. We refer the readers to supplementary material for further details.

**Datasets** We evaluate our method on several commonly used public detection datasets, namely, KITTI [12], PASCAL VOC 2007 [11], MS COCO [6] and ImageNet DET benchmark (ILSVRC 2014) [35]. Among them, KITTI and PASCAL are relatively small datasets that contain less object

| Student | Model Info | Teacher | PASCAL | COCO@.5 | COCO@[.5,.95] | KITTI | ILSVRC |
|---------|-----------|---------|--------|---------|----------------|-------|--------|
| Tucker | 11M / 47ms | - | 54.7 | 25.4 | 11.8 | 49.3 | 20.6 |
| | | AlexNet | 57.6 (+2.9) | 26.5 (+1.2) | 12.3 (+0.5) | 51.4 (+2.1) | 23.6 (+1.3) |
| | | VGGM | 58.2 (+3.5) | 26.4 (+1.1) | 12.2 (+0.4) | 51.4 (+2.1) | 23.9 (+1.6) |
| | | VGG16 | 59.4 (+4.7) | 28.3 (+2.9) | 12.6 (+0.8) | 53.7 (+4.4) | 24.4 (+2.1) |
| AlexNet | 62M / 74ms | - | 57.2 | 32.5 | 15.8 | 55.1 | 27.3 |
| | | VGGM | 59.2 (+2.0) | 33.4 (+0.9) | 16.0 (+0.2) | 56.3 (+1.2) | 28.7 (+1.4) |
| | | VGG16 | 60.1 (+2.9) | 35.8 (+3.3) | 16.9 (+1.1) | 58.3 (+3.2) | 30.1 (+2.8) |
| VGGM | 80M / 86ms | - | 59.8 | 33.6 | 16.1 | 56.7 | 31.1 |
| | | VGG16 | 63.7 (+3.9) | 37.2 (+3.6) | 17.3 (+1.2) | 58.6 (+2.3) | 34.0 (+2.9) |
| VGG16 | 138M / 283ms | - | 70.4 | 45.1 | 24.2 | 59.2 | 35.6 |

Table 1: Comparison of student models associated with different teacher models across four datasets, in terms of mean Average Precision (mAP). Rows with blank (-) teacher indicate the model is without distillation, serving as baselines. The second column reports the number of parameters and speed (per image, on GPU).

categories and labeled images, whereas MS COCO and ILSVRC 2014 are large scale datasets. Since KITTI and ILSVRC 2014 do not provide ground-truth annotation for test sets, we use the training/validation split introduced by [39] and [24] for analysis. For all the datasets, we follow the PASCAL VOC convention to evaluate various models by reporting mean average precision (mAP) at IoU = 0.5 . For MS COCO dataset, besides the PASCAL VOC metric, we also report its own metric, which evaluates mAP averaged for IoU $\in [0.5 : 0.05 : 0.95]$ (denoted as mAP[.5, .95]).

**Models** The teacher and student models defined in our experiments are standard CNN architectures, which consist of regular convolutional layers, fully connected layers, ReLU, dropout layers and softmax layers. We choose several popular CNN architectures as our teacher/student models, namely, AlexNet [27], AlexNet with Tucker Decomposition [26], VGG16 [37] and VGGM [4]. We use two different settings for the student and teacher pairs. In the first set of experiments, we use a smaller network (that is, less parameters) as the student and use a larger one for the teacher (for example, AlexNet as student and VGG16 as teacher). In the second set of experiments, we use smaller input image size for the student model and larger input image size for the teacher, while keeping the network architecture the same.

## 4.1 Overall Performance

Table1 shows mAP for four student models on four object detection databases, with different architectures for teacher guidance. For student models without teacher's supervision, we train them to the best numbers we could achieve. Not surprisingly, larger or deeper models with more parameters perform better than smaller or shallower models, while smaller models run faster than larger ones.

The performance of student models improves significantly with distillation and hint learning over all different pairs and datasets, despite architectural differences between teacher and student. With a fixed scale (number of parameters) of a student model, training from scratch or fine-tuning on its own is not an optimal choice. Getting aid from a better teacher yields larger improvements approaching the teacher's performance. A deeper model as teacher leads to better student performance, which suggests that the knowledge transferred from better teachers is more informative. Notice that the Tucker model trained with VGG16 achieves significantly higher accuracy than the Alexnet in the PASCAL dataset, even though the model size is about 5 times smaller. The observation may support the hypothesis that CNN based object detectors are highly over-parameterized. On the contrary, when the size of dataset is much larger, it becomes much harder to outperform more complicated models. This suggests that it is worth having even higher capacity models for such large scale datasets. Typically, when evaluating efficiency, we get 3 times faster from VGG16 as teacher to AlexNet as student on KITTI dataset. For more detailed runtimes, please refer to supplementary material.

Further, similar to [38], we investigate another student-teacher mode: the student and teacher share exactly the same network structure, while the input for student is down-scaled and the input for teacher remains high resolution. Recent works [1] report that image resolution critically affects object detection performance. On the other hand, downsampling the input size quadratically reduces convolutional resources and speeds up computation. In Table 2, by scaling input sizes to half in

|  | High-res teacher | | Low-res baseline | | Low-res distilled student | |
|---|---|---|---|---|---|---|
|  | mAP | Speed | mAP | Speed | mAP | Speed |
| AlexNet | 57.2 | 1,205 / 74 ms | 53.2 | 726 / 47 ms | 56.7(+3.5) | 726 / 47 ms |
| Tucker | 54.7 | 663 / 41 ms | 48.6 | 430 / 29 ms | 53.5(+4.9) | 430 / 29 ms |

Table 2: Comparison of high-resolution teacher model (trained on images with 688 pixels) and low-resolution student model (trained on 344 pixels input), on PASCAL. We report mAP and speed (both CPU and GPU) of different models. The speed of low-resolution models are about 2 times faster than the corresponding high-resolution models, while achieving almost the same accuracy when our distillation method is used.

| FLOPS(%) | 20 | 25 | 30 | **37.5** | 45 |
|---|---|---|---|---|---|
| Finetune | 30.3 | 49.3 | 51.4 | 54.7 | 55.2 |
| Distillation | 35.5(+5.2) | 55.4(+6.1) | 56.8(+5.4) | 59.4(+4.7) | 59.5(+4.3) |

Table 3: Compressed AlexNet performance evaluated on PASCAL. We compare the model fine-tuned with the ground truth and the model trained with our full method. We vary the compression ratio by FLOPS.

PASCAL VOC dataset for the student and using the original resolution for the teacher, we get almost the same accuracy as the high-resolution teacher while being about two times faster[1].

## 4.2 Speed-Accuracy Trade off in Compressed Models

It is feasible to select CNN models from a wide range of candidates to strike a balance between speed and accuracy. However, off-the-shelf CNN models still may not meet one's computational requirements. Designing new models is one option. But it often requires significant labor towards design and training. More importantly, trained models are often designed for specific tasks, but speed and accuracy trade-offs may change when facing a different task, whereby one may as well train a new model for the new task. In all such situations, distillation becomes an attractive option.

To understand the speed-accuracy trade off in object detection with knowledge distillation, we vary the compression ratio (the ranks of weight matrices) of Alexnet with Tucker decomposition. We measure the compression ratio using FLOPS of the CNN. Experiments in Table 3 show that the accuracy drops dramatically when the network is compressed too much, for example, when compressed size is 20% of original, accuracy drops from 57.2% to only 30.3%. However, for the squeezed networks, our distillation framework is able to recover large amounts of the accuracy drop. For instance, for 37.5% compression, the original squeezed net only achieves 54.7%. In contrast, our proposed method lifts it up to 59.4% with a deep teacher (VGG16), which is even better than the uncompressed AlexNet model 57.2%.

## 4.3 Ablation Study

As shown in Table 4, we compare different strategies for distillation and hint learning to highlight the effectiveness of our proposed novel losses. We choose VGG16 as the teacher model and Tucker as our student model for all the experiments in this section. Other choices reflect similar trends. Recall that proposal classification and bounding box regression are the two main tasks in the Faster-RCNN framework. Traditionally, classification is associated with cross entropy loss, denoted as CLS in Table 4, while bounding box regression is regularized with L2 loss, denoted as L2.

To prevent the classes with small probability being ignored by the objective function, soft label with high temperature, also named weighted cross entropy loss, is proposed for the proposal classification task in Sec.3.2. We compare the weighted cross entropy loss defined in (3), denoted as CLS-W in Table 4, with the standard cross entropy loss (CLS), to achieve slightly better performance on both PASCAL and KITTI datasets.

For bounding box regression, directly parroting to teacher's output will suffer from labeling noise. An improvement is proposed through (4) in Sec.3.3, where the teacher's prediction is used as a boundary to guide the student. Such strategy, denoted as L2-B in Table 4, improves over L2 by 1.3%. Note that a 1% improvement in object detection task is considered very significant, especially on large-scale datasets with voluminous number of images.

| | Baseline | L2 | L2-B | CLS | CLS-W | Hints | Hints-A | L2-B+CLS-W | L2-B+CLS-W+Hints-A |
|---|---|---|---|---|---|---|---|---|---|
| PASCAL | 54.7 | 54.6 | 55.9 | 57.4 | 57.7 | 56.9 | 58 | 58.4 | 59.4 |
| KITTI | 49.3 | 48.5 | 50.1 | 50.8 | 51.3 | 50.3 | 52.1 | 51.7 | 53.7 |

Table 4: The proposed method component comparison, i.e., bounded L2 for regression (L2-B, Sec.3.3) and weighted cross entropy for classification (CLS-W, Sec.3.2) with respect to traditional methods, namely, L2 and cross entropy (CLS). Hints learning w/o adaptation layer (Hints-A and Hints) are also compared. All comparisons take VGG16 as the teacher and Tucker as the student, with evaluations on PASCAL and KITTI.

| | | Baseline | Distillation | Hint | Distillation + Hint |
|---|---|---|---|---|---|
| PASCAL | Trainval | 79.6 | 78.3 | 80.9 | 83.5 |
| | Test | 54.7 | 58.4 | 58 | 59.4 |
| COCO | Train | 45.3 | 45.4 | 47.1 | 49.6 |
| | Val | 25.4 | 26.1 | 27.8 | 28.3 |

Table 5: Performance of distillation and hint learning on different datasets with Tucker and VGG16 pair.

Moreover, we find that the adaptation layer proposed in Sec.3.4 is critical for hint learning. Even if layers from teacher and student models have the same number of neurons, they are almost never in the same feature space. Otherwise, setting the student's subsequent structure to be the same as teacher's, the student would achieve identical results as the teacher. Thus, directly matching a student layer to a teacher layer [2, 3] is unlikely to perform well. Instead, we propose to add an adaptation layer to transfer the student layer feature space to the corresponding teacher layer feature space. Thereby, penalizing the student feature from the teacher feature is better-defined since they lie in the same space, which is supported by the results in Table 4. With adaptation layer, hint learning (Hint-A) shows a 1.1% advantage over the traditional method (Hint). Our proposed overall method (L2-B+CLS-W+Hint-A) outperforms the one without adaptive hint learning (L2-B+CLS-W) by 1.0%, which again suggests the significant advantage of hint learning with adaptation.

## 4.4 Discussion

In this section, we provide further insights into distillation and hint learning. Table 5 compares the accuracy of Tucker model learned with VGG16 on the trainval and testing split of the PASCAL and COCO datasets. In general, distillation mostly improves the generalization capability of student, while hint learning helps improving both the training and testing accuracy.

**Distillation improves generalization:** Similarly to the image classification case discussed in [20], there also exists structural relationship among the labels in object detection task. For example, 'Car' shares more common visual characteristics with 'Truck' than with 'Person'. Such structural information is not available in the ground truth annotations. Thus, injecting such relational information learned with a high capacity teacher model to a student will help generalization capability of the detection model. The result of applying the distillation only shows consistent testing accuracy improvement in Table 5.

**Hint helps both learning and generalization:** We notice that the "under-fitting" is a common problem in object detection even with CNN based models (see low training accuracy of the baselines). Unlike simple classification cases, where it is easy to achieve (near) perfect training accuracy [40], the training accuracy of the detectors is still far from being perfect. It seems the learning algorithm is suffering from the saddle point problem [8]. On the contrary, the hint may provide an effective guidance to avoid the problem by directly having a guidance at an intermediate layer. Thereby, the model learned with hint learning achieves noticeable improvement in both training and testing accuracy.

Finally, by combining both distillation and hint learning, both training and test accuracies are improved significantly compared to the baseline. Table 5 empirically verifies consistent trends on both the PASCAL and MS COCO datasets for object detection. We believe that our methods can also be extended to other tasks that also face similar generalization or under-fitting problems.

## 5 Conclusion

We propose a novel framework for learning compact and fast CNN based object detectors with the knowledge distillation. Highly complicated detector models are used as a teacher to guide the learning process of efficient student models. Combining the knowledge distillation and hint

framework together with our newly proposed loss functions, we demonstrate consistent improvements over various experimental setups. Notably, the compact models trained with our learning framework execute significantly faster than the teachers with almost no accuracy compromises at PASCAL dataset. Our empirical analysis reveals the presence of under-fitting issue in object detector learning, which could provide good insights to further advancement in the field.

**Acknowledgments**    This work was conducted as part of Guobin Chen's internship at NEC Labs America in Cupertino.

## Footnotes

[1]Ideally, the convolutional layers should be about 4 times faster. However, due to the loading overhead and the non-proportional consumption from other layers, this speed up drops to around 2 times faster.

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
