[Reviews · NeurIPS 2017]

Reviewer 1



Overall This paper proposes a framework for training compact and efficient object detection networks using knowledge distillation [20]. It is claimed that this is the first successful application of knowledge distillation to object detection (aside from a concurrent work noted in Section 2 [34]), and to the best of my knowledge I confirm this claim. In my view, the main contributions of this work are the definition of a learning scheme and associated losses necessary to adapt the ideas of [20] and [32] for object detection. This requires some non-trivial problems to be solved - in particular, the definition of the class-weighted cross entropy to handle imbalance between foreground and background classes and the definition of a teacher bounded regression loss to guide the bounding box regression. Another minor contribution is the adaptation layer proposed in 3.4 for hint learning. On a skeptical note, these contributions can be viewed as rather straightforward technical adaptations to apply knowledge distillation to a related but novel task, object detection. In this regard, the goal may be considered a low-hanging fruit and the contributions as incremental. I have two main criticisms to the manuscript. First, I am not convinced that knowledge distillation is particularly well suited to the problem of object detection (see argument below). This leads to my second concern: there are no experiments demonstrating advantages of the proposed model compression method to other model compression approaches. The manuscript is well written and easy to read, with only a few minor typos. The related work is well structured and complete, and the experiments seem fairly rigorous. Related Works + The related works is well structured and seems to be comprehensive. The related work section even identifies a recent concurrent work with on the same topic and clearly points out the difference in the two approaches. Approach - It seems to me that knowledge distillation is not particularly well suited to the problem of object detection where there are only a few classes (and quite often, only foreground and background). One of the main features of [20] is the ability to distill knowledge from an ensemble containing generalist and specialist models into a single streamlined model. The specialist models should handle rare classes or classes that are easily confused, usually subsets of a larger class (such as dog breeds). There is no use of ensemble learning in the proposed approach, and there would be little benefit as there are relatively few classes. My intuition is that this will give much less power to the soft targets and knowledge distillation. - The problem of class balancing for object detection needs to be more clearly motivated. What are the detection rates from the RPN and the ratio of foreground to background? Experiments + Good experimental design clearly demonstrates the advantages of the method. + Important commonly used benchmark datasets are used (PASCAL, MS COCO, ILSVRC) + Experiments use several teacher and student models in a variety of configurations, allowing the reader to clearly identify trends (deeper teachers = better performance, bigger datasets = smaller gains) + Ablation study justifies the choice of weighted cross-entropy and bounded regression loss functions (note: In my opinion ‘ablation’ is a slight misuse of the terminology, although it is used this way in other works) - A comparison against other compressed models is lacking. - It would have been interesting to see results with a deeper and more modern network than VGG-16 as teacher. Other comments L29 - Deeper networks are easier to train - clarify what you mean by this. Problems like vanishing gradients make it difficult to train deeper networks. L40-41 - This line is unclear. Should you replace ‘degrades more rapidly’ with ‘suffers more degradation’? L108 - Typos: features, takes L222 - Typo: Label 1

Reviewer 2



Paper Summary: The idea of the paper is to learn compact models for object detection using knowledge distillation and hint learning. In knowledge distillation, a student model (the compact model) learns to predict the probability distribution that the teacher model (the larger more powerful network) predicts (as opposed to the groundtruth labels). In hint learning, the goal is to learn the intermediate layers of the teacher network. The model is a modification of Faster-RCNN that incorporates new losses for knowledge distillation and hint learning. Strengths: - The results are quite interesting that compact models are almost as good as the deeper and larger networks. - The paper provides thorough analysis of different networks and different datasets. - The paper is the first paper that uses knowledge distillation for object detection. - The ablation study is very informative and shows how different components of the model help improving the performance. Weaknesses: - The paper is a bit incremental. Basically, knowledge distillation is applied to object detection (as opposed to classification as in the original paper). - Table 4 is incomplete. It should include the results for all four datasets. - In the related work section, the class of binary networks is missing. These networks are also efficient and compact. Example papers are: * XNOR-Net: ImageNet Classification Using Binary Convolutional Neural Networks, ECCV 2016 * Binaryconnect: Training deep neural networks with binary weights during propagations, NIPS 2015 Overall assessment: The idea of the paper is interesting. The experiment section is solid. Hence, I recommend acceptance of the paper.

Reviewer 3



The paper presents a very nice idea introducing knowledge distillation to object class detection in images. It has several positive and negative aspects. Positive aspects: 1. Application oriented. The paper is concerned with several practical aspects related to object detection, the main being keeping decent accuracy at lower computational cost. This is very interesting from practicioners point of view. 2. The paper is well written and easy to read. 3. The paper reffers to relevant object detection and knowledge distillation literature fairly and adequately. 4. The experiments suggest that the proposed method can indeed achieve reasonable performance (sligntly bellow the uncompressed models) at a cheaper computational cost accross 4 different object detection datasets. 5. The paper provides analysis and insights into the different aspects and novelties introduced in this work. Negative aspects: 1. Novelty. The idea of knowledge distilation is not new and has been explored before in the literature. The novelty in this work comes via several engineering tricks the paper employs to adapt the knowledge distilation idea to the Faster-RCNN framework. 2. Comparison to baselines and state-of-the-art. The paper failed to compare to relevant and newer state-of-the-art works on object detection outside the Faster-RCNN framework. In addition, comparison to other compression and efficiency-oriented works would be good to have, both in terms of quality but also speed. 3. Limited experiments to AlexNet and VGG. It would be great to see if the method scales to bigger models like Inception and ResNet. In addition, it would be interesting to see if using a smaller model as teacher helps the bigger student model. 4. In the end, it seems distilation is a technique to get better local minimum when training models. Detailed comments: 1. Line 44: The claim that object class detection is more complex than image classification is an overstatement. 2. Equation 1 should clearly state what is new w.r.t. Faster-RCNN, if any at all. 3. Line 127. How do you fix the lambda and gamma params? 4. Eq 2. How do you fix the mu param? 5. Line 148. How do you fix the weights? 6. Eq 6. What about m? 7. Table 3 seems unfair in two dimensions. First, the finetuned model is compressed, but then the distillation model is trained as VGG16 as teacher. It would be good to keep the teacher to be the uncompressed student model (alexnet). This will reduce the degrees of freedom in the experiment.